# The N-Methyl-D-Aspartate Receptor Blocker REL-1017 (Esmethadone) Reduces Calcium Influx Induced by Glutamate, Quinolinic Acid, and Gentamicin

**DOI:** 10.3390/ph15070882

**Published:** 2022-07-17

**Authors:** Ezio Bettini, Sara De Martin, Andrea Mattarei, Marco Pappagallo, Stephen M. Stahl, Francesco Bifari, Charles E. Inturrisi, Franco Folli, Sergio Traversa, Paolo L. Manfredi

**Affiliations:** 1In Vitro Pharmacology Department, Aptuit, an Evotec Company, 37135 Verona, Italy; ezio.bettini@evotec.com; 2Department of Pharmaceutical and Pharmacological Sciences, University of Padua, 35122 Padua, Italy; sara.demartin@unipd.it (S.D.M.); andrea.mattarei@unipd.it (A.M.); 3Relmada Therapeutics, Inc., Coral Gables, FL 33134, USA; mpappagallo@relmada.com (M.P.); ceintur@gmail.com (C.E.I.); st@relmada.com (S.T.); 4Department of Anesthesiology, Albert Einstein College of Medicine, Bronx, NY 10461, USA; 5Department of Psychiatry, University of California, San Diego, La Jolla, CA 92093, USA; smstahl@neiglobal.com; 6Neuroscience Education Institute, Carlsbad, CA 92008, USA; 7Department of Medical Biotechnology and Translational Medicine, University of Milan, 20122 Milan, Italy; francesco.bifari@unimi.it; 8Department of Health Sciences, University of Milan, 20122 Milan, Italy; franco.folli@unimi.it

**Keywords:** COVID-19, esmethadone, gentamicin, major depressive disorder (MDD), neuropsychiatric, N-methyl-D-aspartate receptor, quinolinic acid, REL-1017

## Abstract

REL-1017 (esmethadone) is a novel N-methyl-D-aspartate receptor (NMDAR) antagonist and promising rapid antidepressant candidate. Using fluorometric imaging plate reader (FLIPR) assays, we studied the effects of quinolinic acid (QA) and gentamicin, with or without L-glutamate and REL-1017, on intracellular calcium ([Ca^2+^]_in_) in recombinant cell lines expressing human GluN1-GluN2A, GluN1-GluN2B, GluN1-GluN2C, and GluN1-GluN2D NMDAR subtypes. There were no effects of QA on [Ca^2+^]_in_ in cells expressing GluN1-GluN2C subtypes. QA acted as a low-potency, subtype-selective, NMDAR partial agonist in GluN1-GluN2A, GluN1-GluN2B, and GluN1-GluN2D subtypes. REL-1017 reduced [Ca^2+^]_in_ induced by QA. In cells expressing the GluN1-GluN2D subtype, QA acted as an agonist in the presence of 0.04 μM L-glutamate and as an antagonist in the presence of 0.2 μM L-glutamate. REL-1017 reduced [Ca^2+^]_in_ induced by L-glutamate alone and with QA in all cell lines. In the absence of L-glutamate, gentamicin had no effect. Gentamicin was a positive modulator for GluN1-GluN2B subtypes at 10 μM L-glutamate, for GluN1-GluN2A at 0.2 μM L-glutamate, and for GluN1-GluN2A, GluN1-GluN2B, and GluN1-GluN2D at 0.04 μM L-glutamate. No significant changes were observed with GluN1-GluN2C NMDARs. REL-1017 reduced [Ca^2+^]_in_ induced by the addition of L-glutamate in all NMDAR cell lines in the presence or absence of gentamicin. In conclusion, REL-1017 reduced [Ca^2+^]_in_ induced by L-glutamate alone and when increased by QA and gentamicin. REL-1017 may protect cells from excessive calcium entry via NMDARs hyperactivated by endogenous and exogenous molecules.

## 1. Introduction

REL-1017 (esmethadone; dextromethadone; d-methadone; S-methadone; (+)-methadone) is the opioid-inactive d-isomer of racemic methadone and a low-affinity, low-potency N-methyl-D-aspartate receptor (NMDAR) channel blocker that binds to the MK-801 site of the NMDAR at low micromolar half-maximal inhibitory concentration (IC_50_) values [1,2,3]. REL-1017 showed efficacy in animal models of depressive-like behavior [4,5] and rapid, robust, and sustained therapeutic effects as adjunctive treatment in patients with major depressive disorder (MDD) [6]. The clinical findings in patients also confirmed the favorable safety, tolerability, and pharmacokinetic profiles observed in Phase 1 studies [7]. NMDAR antagonists cause neuronal vacuoles, known as Olney’s lesions, in rats, which may be related to NMDAR toxicity in humans [8]. In a recent experiment, REL-1017 did not cause Olney’s lesions, in contrast with MK-801 [9]. In summary, these studies suggest that REL-1017 is a safe and well-tolerated novel NMDAR channel blocker and rapid antidepressant candidate. To gain additional insights into the mechanism of action of REL-1017, we investigated its effects by means of fluorometric imaging plate reader (FLIPR) assays in the presence of putative NMDAR function-associated endogenous and exogenous neurotoxins, such as quinolinic acid (QA) and gentamicin.

QA is a 2,3-pyridine dicarboxylic acid produced by the breakdown of the amino acid tryptophan via the kynurenine pathway. QA is physiologically present in nanomolar concentrations in the brain; however, increased levels of QA can be produced by activated macrophages and microglia in pathological conditions. Accumulation of endogenous QA has been implicated in the etiology of neurological diseases [10,11] and psychiatric disorders, including depression [12,13,14,15,16,17,18,19]. QA has also been implicated in peripheral neurodegenerative disorders, such as hearing loss [20,21], and has recently been implicated in the neuropsychiatric complications of COVID-19 [22,23]. A variety of mechanisms have been proposed to explain QA toxicity, including activation of NMDARs via an increase in ambient glutamate and via direct agonist actions at the glutamate site of the NMDAR [11,24,25,26,27,28]. Of note, QA-induced lesions differ across multiple brain regions, suggesting that QA could exert regional effects based on the specific expression of NMDAR subtypes at the site of the lesion [29]. Electrophysiological recording in oocytes overexpressing recombinant NMDARs provided direct evidence of NMDAR activation by QA, with low-potency partial agonist activity at L-glutamate binding sites on GluN1-GluN2A, GluN1-GluN2B, GluN1-GluN2D, and GluN1-GluN3B NMDAR subtypes [30,31,32,33,34]. QA has been shown to be inactive at GluN1-GluN2C subtypes [30]. QA activation of NMDARs may also have a causative role in peripheral diseases [35]. QA has been shown to accumulate during liver failure [36] and renal insufficiency [37,38].

The current study utilized QA to activate NMDARs in vitro in order to better evaluate the role of REL-1017 in modulating pathologically hyperactive NMDARs. We first verified if QA could modulate NMDAR-mediated intracellular calcium ([Ca^2+^]_in_) in recombinant cell lines expressing heterodimeric NMDARs with subtype specificity. We then assessed the ability of REL-1017 to antagonize QA effects on NMDARs, alone and in the presence of glutamate.

In order to further assess the role of REL-1017 in modulating NMDAR activity, REL-1017 was evaluated in the presence of the aminoglycoside antibiotic gentamicin. Gentamicin acts mainly on aerobic Gram-negative bacteria, including multidrug-resistant bacteria. However, its clinical usefulness is limited by potentially serious adverse effects, most commonly ototoxicity, including both cochlear and vestibular toxicity [39,40], and nephrotoxicity [39,41]. A variety of mechanisms have been proposed to explain aminoglycoside side effects [39,40], including activation of NMDARs [42,43,44,45], which are expressed in various cell types of the inner ear [46] and kidney [47]. Direct evidence supports activation of NMDARs by gentamicin [45,48,49,50] and tobramycin, another aminoglycoside antibiotic [48,51]. Therefore, we also directly tested if gentamicin could increase NMDAR-mediated [Ca^2+^]_in_ in recombinant cell lines expressing heterodimeric NMDARs, both in the presence and absence of glutamate. We then assessed the ability of REL-1017 to antagonize the effects of gentamicin on NMDARs activated by glutamate and the ability of REL-1017 to antagonize the effects of gentamicin and QA.

These experiments may improve our understanding of the mechanism of action of REL-1017, including its potential to protect cells from excessive calcium entry via NMDARs hyperactivated by glutamate and endogenous or exogenous substances, such as QA and gentamicin.

## 2. Results

### 2.1. QA Acts as a Low-Potency Subtype-Selective NMDAR Partial Agonist, and REL-1017 Reduces QA-Induced Calcium Entry

The effect of QA on NMDAR subtypes is shown in Figure 1A and Table 1. An increase in [Ca^2+^]_in_ was detected with the addition of QA and in the absence of L-glutamate in GluN1-GluN2A, GluN1-GluN2B, and GluN1-GluN2D cell lines. In addition, 333 µM QA enhanced [Ca^2+^]_in_ by an average of 18%, 23%, and 21% in cells expressing GluN2A, GluN2B, and GluN2D, respectively (*n* = 6 for each group), compared to a 100% value obtained with 10 µM L-glutamate. At the highest tested concentration (1000 µM), QA enhanced [Ca^2+^]_in_ by an average of 40%, 25%, and 50% of the maximum response to L-glutamate alone (*n* = 6 for each group) in cells expressing GluN2A, GluN2B, and GluN2D, respectively. QA was inactive on the GluN1-GluN2C cell line at all tested concentrations. Therefore, in this FLIPR calcium assay, QA acted as a low-potency subtype-selective NMDAR partial agonist.

We then tested the effect of 10 µM REL-1017 on the increases in [Ca^2+^]_in_ induced by 1000 µM QA. The results confirmed that 1000 μM QA induced increases in [Ca^2+^]_in_ in GluN1-GluN2A-, GluN1-GluN2B-, and GluN1-GluN2D-expressing cells by an average of 41%, 37%, and 55% (*n* = 42 for each group), respectively, compared to a 100% value obtained with 10 µM L-glutamate (Figure 1B, Table 1). In contrast, the 1000 μM QA-induced [Ca^2+^]_in_ in GluN1-GluN2C-expressing cells was −5.5% (*n* = 42), which was slightly, but significantly, lower compared to calcium levels in the presence of the buffer alone (*p* < 0.05). This suggests the presence of trace L-glutamate released in the buffer by Chinese hamster ovary (CHO) in basal conditions and an antagonist effect of 1000 μM QA. REL-1017 10 μM significantly (*p* < 0.0001) modulated [Ca^2+^]_in_ in all four cell lines by an average of 34%, 12%, −7.6%, and 32% (*n* = 42 for each group) for GluN1-GluN2A, GluN1-GluN2B, GluN1-GluN2C, and GluN1-GluN2D subtypes, respectively.

### 2.2. REL-1017 Reduces Calcium Entry in the Presence of 0.04 μM or 0.2 μM L-Glutamate Plus 1000 μM QA

In order to verify and characterize the possible interactions between QA and low L-glutamate concentrations with [Ca^2+^]_in_, we treated cells with 0.04 μM or 0.2 μM L-glutamate in either the presence or absence of 1000 μM QA (Figure 2A). Consistent with the results obtained for the concentration–response curve, 1000 μM QA significantly (*p* < 0.0001) potentiated [Ca^2+^]_in_ induced by either 0.04 μM or 0.2 μM L-glutamate in cells expressing GluN1-GluN2A and GluN1-GluN2B subtypes (Figure 2A,B; Table 2). In cells expressing the GluN1-GluN2C subtype, 1000 μM QA did not change the calcium levels induced by 0.04 μM L-glutamate (Figure 2A, *p* = 0.5041) and reduced [Ca^2+^]_in_ induced by 0.2 μM L-glutamate (Figure 2B, *p* < 0.0001). In the cell lines expressing the GluN1-GluN2D subtype, 1000 μM QA significantly increased [Ca^2+^]_in_ induced by 0.04 μM L-glutamate (Figure 2A, *p* < 0.0001), while it significantly reduced the [Ca^2+^]_in_ induced by 0.2 μM L-glutamate (Figure 2B, *p* < 0.0001). Thus, QA acted as an agonist in the presence of 0.04 μM L-glutamate and as an antagonist in the presence of 0.2 μM L-glutamate in the cell lines expressing the GluN1-GluN2D subtype. When 10 μM REL-1017 was added to 1000 μM QA in the presence of 0.04 μM or 0.2 μM L-glutamate, [Ca^2+^]_in_ was significantly (*p* < 0.0001) reduced in all four cells lines (Figure 2A,B; Table 3).

### 2.3. Gentamicin Is a Selective Positive Modulator at NMDAR GluN1-GluN2B Subtypes for Saturating L-Glutamate Concentration

We tested if gentamicin could activate NMDARs by acting as an agonist in the absence of L-glutamate. The ability of gentamicin to increase [Ca^2+^]_in_ was first assessed in the absence of L-glutamate but in the presence of 10 μM glycine. A gentamicin CRC using eleven different concentrations of gentamicin ranging from 1.7 ng/mL up to 100 μg/mL was tested in a FLIPR calcium assay using four CHO cell lines, each expressing a different heterodimeric NMDAR, including GluN1-GluN2A, GluN1-GluN2B, GluN1-GluN2C, and GluN1-GluN2D. No increase in [Ca^2+^]_in_ was detected with the addition of gentamicin in the absence of L-glutamate at any concentration and in any cell line (Figure 3A). At the tested concentrations, gentamicin had no effect as an agonist on NMDAR by substituting L-glutamate.

We tested if gentamicin could act as a positive modulator on NMDARs by enhancing the L-glutamate-induced maximal calcium response. A FLIPR calcium assay was designed to evaluate changes to the gentamicin CRC with the addition of 10 μM L-glutamate, using the same NMDAR cell lines used for the agonist assay. In the cell line expressing the GluN1-GluN2B NMDAR subtype, 100 μg/mL gentamicin significantly (*p* < 0.0001, unpaired *t*-test versus control) increased [Ca^2+^]_in_ compared to levels elicited by 10 μM L-glutamate alone (Figure 3B). The increase with added gentamicin elicited a 128% change in [Ca^2+^]_in_ relative to the maximum response recorded in the presence of 10 μM L-glutamate alone. The calculated EC_50_ of gentamicin for the GluN1-GluN2B subtype was 21 μM. Considering that [Ca^2+^]_in_ reaches a plateau following induction by 10 μM L-glutamate, the further change to [Ca^2+^]_in_ induced by the additional gentamicin can be considered a greater than maximal response on GluN1-GluN2B NMDARs. In contrast, 100 μg/mL gentamicin induced a significant (*p* < 0.0001) reduction in [Ca^2+^]_in_ of 80%, 65%, and 78% in the remaining cell lines of GluN1-GluN2A, GluN1-GluN2C, and GluN1-GluN2D subtypes, respectively, acting as a negative allosteric modulator on NMDAR function in these cell lines exposed to 10 μM L-glutamate.

### 2.4. Gentamicin Increases NMDAR Affinity for Glutamate at Sub-Saturating L-Glutamate Concentrations in a Glutamate Concentration- and GluN1-GluN2 Subtype-Dependent Manner

We then tested if gentamicin could modulate the effects of submaximal concentrations of L-glutamate (Figure 4A; Table 4). The effect of 10 μg/mL gentamicin on [Ca^2+^]_in_ was evaluated in the presence of 0.04 μM L-glutamate, which alone produced changes in [Ca^2+^]_in_ relative to the 10 μg/mL gentamicin effect of 5.4%, 6.6%, 19%, and 26% for GluN1-GluN2A, -GluN2B, -GluN2C, and -GluN2D NMDARs, respectively (Figure 4A). We also tested the effect of 10 μg/mL gentamicin in the presence of 0.2 μM L-glutamate, which alone produced a change in [Ca^2+^]_in_ of 30%, 47%, 66%, and 87%, with GluN1-GluN2A, -GluN2B, -GluN2C, and -GluN2D NMDARs, respectively (Figure 4B; Table 5). In addition, 10 μg/mL gentamicin acted as a positive modulator in the presence of 0.04 μM L-glutamate, not only for GluN1-GluN2B NMDARs as expected (*p* < 0.0001), but also for GluN1-GluN2A and GluN1-GluN2D NMDARs (*p* < 0.0001 and *p* < 0.001, respectively), while no significant changes were observed with GluN1-GluN2C NMDARs (Figure 4A; Table 4). It was also shown that 10 μg/mL gentamicin acted as a positive modulator in the presence of 0.2 μM L-glutamate with only GluN1-GluN2A NMDARs, increasing [Ca^2+^]_in_ to 37% of the maximum response (*p* < 0.0001), while no significant changes occurred with the remaining NMDARs (Figure 4B; Table 5). REL-1017 reduced [Ca^2+^]_in_ induced by the addition of 0.04 and 0.02 mM glutamate, also in the absence of gentamicin. Glutamate acts on NMDARs, increasing calcium currents, and REL-1017 is an NMDAR blocker, and therefore decreases calcium currents.

### 2.5. Gentamicin Is Not a Positive Modulator at NMDARs Activated by Quinolinic Acid

We then tested the effect of gentamicin on QA-induced changes to [Ca^2+^]_in_, as it is known that positive modulator effects might be agonist dependent. Additionally, QA is an endogenous metabolite that acts as an agonist at the glutamate site of the NMDAR and has neurotoxic and ototoxic potential [21]. QA is a weak, subtype-selective, partial agonist at the L-glutamate site of NMDARs; 1000 μM QA induced changes in [Ca^2+^]_in_ of 42%, 37%, and 55% of the 10 µM L-glutamate response at GluN1-GluN2A, -GluN2B, and -GluN2D NMDARs (*p* < 0.0001), respectively, and was inactive on GluN1-GluN2C subtypes (Figure 5; Table 6). When tested in combination with 1000 μM QA, 10 μg/mL gentamicin was inactive on all tested NMDARs.

### 2.6. REL-1017 Lowered Intracellular Calcium Levels in Presence of 10 μM Gentamicin and Low L-Glutamate

After showing that REL-1017 reduced [Ca^2+^]_in_ that was increased after NMDAR activation by QA (Figure 1), we investigated if REL-1017 was also effective in reducing [Ca^2+^]_in_ increased by gentamicin, a positive modulator at the NMDAR, in the presence of low glutamate concentrations. It was found that 10 μM REL-1017 reduced [Ca^2+^]_in_ induced by the addition of 0.04 or 0.02 µM L-glutamate in all NMDAR cell lines, in the presence or absence of 10 μg/mL gentamicin (Figure 4, *p* < 0.0001 or *p* < 0.001). In addition, 10 μM REL-1017 also reduced [Ca^2+^]_in_ elicited by 1000 μM QA in the presence and absence of 10 μg/mL gentamicin in all NMDAR cell lines (Figure 5, *p* < 0.0001).

## 3. Discussion

The central role of NMDARs in neural plasticity has been recognized for several decades [52,53]. The distribution of NMDAR subtypes is genetically, regionally, and developmentally determined [53]. Diseases may be triggered, maintained, or worsened by dysregulated calcium currents through NMDARs [54], including peripheral NMDARs [55]. The effect of QA on NMDAR in this intracellular FLIPR calcium assay confirms and extends the data obtained by electrophysiological recordings in transduced oocytes expressing heterodimeric NMDARs [30,31,32,33,34]. In particular, we confirmed that QA is a weak partial agonist at GluN1-GluN2A, GluN1-GluN2B, and GluN1-GluN2D subtypes. QA significantly increased [Ca^2+^]_in_ only at the two highest tested concentrations, 333 μM and 1000 μM (Figure 1). QA alone was inactive at every tested concentration up to 1000 μM on heterodimeric GluN1-GluN2C subtypes (Figure 1), as previously described [30]. Our results support the hypothesis that brain region-specific patterns of QA-induced lesions may be caused by preferential agonist activity at regionally and developmentally specific NMDAR subtypes [29].

We obtained additional information regarding the QA interactions with NMDARs by studying the effect of 1000 µM QA added to cell culture media containing low L-glutamate concentrations. It was found that 1000 μM QA potentiated the effect of 0.04 μM and 0.2 μM L-glutamate on GluN1-GluN2A or GluN1-GluN2B NMDARs (Figure 2). This potentiation was observed with L-glutamate concentrations well below the EC_50_ for NMDAR heterodimeric GluN1-GluN2A or GluN1-GluN2B subtypes. In addition, 1000 µM QA is expected instead to decrease the response to saturating L-glutamate concentrations on GluN1-GluN2A or GluN1-GluN2B, but this condition remains to be tested. Divergent effects of 1000 μM QA were instead observed with [Ca^2+^]_in_ elicited by 0.04 μM and 0.2 μM L-glutamate at the GluN1-GluN2D subtype. Approximately 18% of the maximum [Ca^2+^]_in_ induction was elicited by 0.04 μM L-glutamate alone, which increased to 55% in the presence of 1000 μM QA (Figure 2A, *p* < 0.0001). In contrast, an average of 92% of the maximum [Ca^2+^]_in_ induction was elicited by 0.2 μΜ L-glutamate alone, while the combination with 1000 μM QA significantly decreased the 0.2 μΜ L-glutamate effect on [Ca^2+^]_in_ response in GluN1-GluN2D heterodimeric NMDARs (Figure 2B, *p* < 0.0001). Interestingly, 0.04 μM and 0.2 μM L-glutamate concentrations represent values below and above the L-glutamate EC_50_ for the GluN1-GluN2D subtype in this assay, respectively; this observation may help explain the contrasting outcomes. Moreover, 1000 μM QA decreased [Ca^2+^]_in_ elicited by 0.2 μM L-glutamate in GluN1-GluN2C NMDAR subtypes (*p* < 0.0001; Figure 2B). This result suggests that QA can bind to the L-glutamate site of GluN1-GluN2C subtypes and compete with L-glutamate without a pharmacological effect on this receptor subtype, as QA alone is unable to activate the GluN1-GluN2C NMDAR subtypes.

Considering that sub-micromolar concentrations of ambient glutamate tonically activate NMDARs in select cell types and/or brain regions, physiologically and pathologically [54,56,57,58,59], our observed QA potentiating effect of sub-saturating L-glutamate concentrations may help explain the quinolinic-induced neurotoxicity by GluN1-GluN2A, GluN1-GluN2B, or GluN1-GluN2D NMDAR subtype hyperactivity (Figure 2). With regard to the GluN1-GluN2D subtypes, potentiation was evident only at 0.04 μM and not at 0.2 μM, suggesting that QA-induced calcium dysregulation at this receptor subtype may be preferentially relevant in the presence of very low ambient glutamate. As ambient glutamate in physiological states is expected to be negligible, the importance of this receptor subtype as the most vulnerable target in the pathophysiology of disorders triggered or maintained by excess ambient glutamate, even at very low concentrations, may deserve more attention. Of note, this receptor subtype is the preferential target for REL-1017 in the presence of physiological magnesium [60]. Finally, the effect of REL-1017 on the reduction in [Ca^2+^]_in_ induced by QA, alone (Figure 1) or in combination with sub-saturating L-glutamate concentrations (Figure 2), suggests potential neuroprotective effects when QA might overstimulate NMDARs directly, alone and in the presence of low-concentration ambient glutamate.

The effects of gentamicin on NMDARs in our intracellular calcium assay confirm and extend the data obtained through electrophysiological recordings from transduced oocytes expressing heterodimeric NMDARs [48,49,50]. In particular, we confirmed the selectivity of gentamicin for the GluN1-GluN2B receptor at 10 µM L-glutamate concentration (Figure 3), with an EC_50_ of 21 µg/mL (30.8 µM). Exposure to 100 µg/mL (146.7 µM) gentamicin in addition to 10 µM L-glutamate induced a [Ca^2+^]_in_ increase of 128% of the previously measured plateau from saturating L-glutamate concentrations alone. This finding is in agreement with the 38 µM EC_50_ and 132% plateau maximal response with 300 µM gentamicin described in oocytes [48]. In addition, we observed a potentiation by 10 µg/mL gentamicin of the sub-saturating 0.04 µM glutamate effect on GluN1-GluN2A, GluN1-GluN2B, and GluN1-GluN2D NMDAR subtypes (Figure 4A), while potentiation of a slightly higher but still sub-saturating 0.2 µM glutamate effect was observed only with GluN1-GluN2A NMDAR subtypes (Figure 4B).

Recently, NMDAR positive allosteric modulators (PAMs) have been grouped into the following two classes [61]: type I, which modulates the maximal glutamate effect, and type II, which increases agonist affinity. Our reported results suggest that gentamicin can be considered a selective GluN1-GluN2B type I PAM. Further studies should be performed to establish if gentamicin could also be considered a GluN2A, GluN2B, or GluN2D type II PAM. Considering that sub-micromolar concentrations of ambient glutamate might tonically activate NMDARs in regional- and developmentally-specific neuronal populations [57], the observed gentamicin potentiating effect of sub-saturating L-glutamate concentrations might indicate a previously unappreciated mechanism of gentamicin neurotoxicity. The effect of REL-1017 in reducing [Ca^2+^]_in_ induced by 10 µg/mL gentamicin and sub-saturating L-glutamate concentrations suggests the potential neuroprotective effects of REL-1017 against gentamicin-associated NMDAR excitotoxicity in the context of low ambient glutamate.

REL-1017 might, therefore, provide neuroprotective channel blocking effects in conditions of NMDAR hyperactivation from potential endogenous and exogenous neurotoxins. REL-1017 is currently in Phase 3 trials for the treatment of MDD (ClinicalTrials.gov Identifiers: NCT04855747; NCT04688164; NCT05081167) and is under investigation for the treatment of other neuropsychiatric and neurodegenerative disorders. Dysregulated NMDAR-mediated calcium currents have been implicated in the pathophysiology of neuropsychiatric and central neurodegenerative disorders, in addition to peripheral disorders [62], including COVID-19 [22,23]. The genetic and environmental paradigm, stating that genetically predisposed individuals exposed to select triggers may develop diseases, is generally accepted [63]. QA is an endogenous substance with the potential to elicit neurotoxicity via excessive NMDAR activity, which has also been implicated in the development of neurodegenerative disorders. Gentamicin is an antibiotic with toxicity, which is also potentially mediated via NMDAR hyperactivation.

Uncompetitive NMDAR channel blockers have been proven effective for the treatment of neuropsychiatric diseases. Memantine, esketamine, and the combination dextromethorphan-quinidine are FDA approved for Alzheimer’s disease, treatment-resistant major depressive disorder, and for pseudobulbar palsy, respectively.

Improvements in the knowledge and characterization of NMDARs’ function, their agonists, their allosteric modulators, and their antagonists, combined with the recent availability of clinical data for NMDAR-modulating drugs, are likely to progress our understanding of NMDAR functions in health and disease and are likely to improve our ability to understand and treat diseases associated with this receptor system. The results of this study suggest that REL-1017 could be a novel treatment for diseases triggered, maintained, or worsened by excessive NMDAR-dependent calcium currents, particularly via GluN1-GluN2D subtypes.

## 4. Materials and Methods

### 4.1. Drugs and Reagents

All chemicals were of analytical grade. The following chemicals were from Merck Sigma-Aldrich (Milan, Italy): QA (Cat # P63204), L-glutamate (G1626), glycine (G7403), and probenecid (Cat # P8761). Ketamine was from Merial (Cat # Imalgene 1000). Esmethadone hydrochloride (CAS: 15284-15-8) was from SpecGx (Webster Groves, MO, USA; Cat # 8514). Fluo-4 (F14202) was from Invitrogen (Milan, Italy). The following cell culture chemicals were from Gibco Life Technologies (Milan, Italy): Dulbecco’s Modified Eagle Medium/F12 (DMEM/F12; Cat # 21331), heat-inactivated fetal bovine serum (Cat # 10500), penicillin-streptomycin (Cat # 15140), L-glutamine (Cat # 25030), MEM nonessential amino acids (Cat # 11140), blasticidin (Cat # A11139), G418 (Cat # 10131), Zeocin™ (Cat # R250), and TrypLE™ enzyme (Cat # 12604). Hygromycin B (Cat # 10687) was from Invitrogen. Gentamicin sulphate concentrations were expressed in μg/mL as gentamicin sulphate was a mixture of three major components designated as C1, C1a, and C2. Calculated and corrected for purity, the molecular weight was 681.58 Da, such that 100 µg/mL corresponded to 146.7 µM gentamicin sulphate.

### 4.2. Cell Lines

CHO cell lines stably expressing heterodimeric GluN1-GluN2A, GluN1-GluN2B, GluN1-GluN2C, and GluN1-GluN2D were produced by Aptuit, an Evotec company (Verona, Italy). Research resource identifiers (RRIDs) for the cell lines used were CVCL_B3TY, CVCL_B3TZ, CVCL_B3U0, and CVCL_B3U1 for CHO-hGluN1-hGluN2A, CHO-hGluN1-hGluN2B, CHO-hGluN1-hGluN2C, and CHO-hGluN1-hGluN2D. None of the cell lines used were listed as commonly misidentified cell lines by the International Cell Line Authentication Committee. Cell lines were pharmacologically characterized in 2018 [60]. Protein accession numbers were NP_015566, NP_000824, NP_000825, NP_000826, and NP_000827 for GluN1, GluN2A, GluN2B, GluN2C, and GluN2D, respectively. The four different heterodimeric NMDARs studied were GluN1-GluN2A, GluN1-GluN2B, GluN1-GluN2C, and GluN1-GluN2D. Every cell line was used for a maximum of 75 passages. Cells were grown at 37 °C, 5% CO_2_ in DMEM/F12, with 10% FBS, 1% penicillin-streptomycin, 2 mM L-glutamine, 1% MEM-NEAA, 500 µM ketamine, and antibiotics. Antibiotics were 10 µg/mL blasticidin, 400 µg/mL G418, 400 µg/mL hygromycin B (except for hGluN1/hGluN2B-CHO), and 300 µg/mL Zeocin™ (for hGluN1/hGluN2B-CHO only).

### 4.3. FLIPR Assay

For the FLIPR (Molecular Devices, San Jose, CA, USA) assay, cells were plated in 384 black clear-bottom plates for 24 h, at a density of 15,000/well, in the presence of 500 µM ketamine and 10 µg/mL tetracycline to induce receptor expression (except for GluN1/GluN2B-CHO, which showed constitutive NMDAR expression). Plated cells were preloaded for 1 h with 2 μM Fluo-4 calcium sensitive dye in the presence of 2.5 mM probenecid, an inhibitor of nonspecific anion transport, and 500 μM ketamine, to avoid cytotoxicity, and then washed with assay buffer. Assay buffer composition was 145 mM NaCl, 5 mM KCl, 2 mM CaCl_2_, 1 mg/mL D-(+)-glucose, and 20 mM HEPES (pH adjusted to 7.3 with NaOH).

[Ca^2+^]_in_ was monitored by FLIPR (excitation wavelength at 470-495 nm, emission wavelength at 515-575 nm) for 10 s before and 5 min after the addition of the test items. All test items, including L-glutamate, when present, and glycine, were added simultaneously. In addition, 10 μM glycine was included in every test item addition. In FLIPR CRC experiments, QA was tested at 11 final concentrations, which were as follows: 1000 μM, 330 μM, 110 μM, 37 μM, 12 μM, −4.12 μM, and −1.37 μM, then 460 nM, 150 nM, 51 nM, and 17 nM. In FLIPR CRC experiments, gentamicin was assessed at 11 final concentrations, which were as follows: 100 μg/mL, 33 μg/mL, 11 μg/mL, 3.7 μg/mL, and 1.2 μg/mL, then 412 ng/mL, 137 ng/mL, 46 ng/mL, 15 ng/mL, 5.1 ng/mL, and 1.7 ng/mL.

### 4.4. Statistical Analysis

The study was not preregistered, and Institutional Ethics Committee approval was not necessary as the studies were performed on cell lines. Sample size was determined based on similar cell culture studies [64]. As these were in vitro cell culture experiments, no randomization procedures were applied and no blinding, sample calculation, or test for outliers were performed. 

FLIPR data were expressed as a percentage of response, considering the [Ca^2+^]_in_ in the presence of 10 μM L-glutamate and 10 μM glycine as the 100% response (H, high control) and [Ca^2+^]_in_ in assay buffer alone as the 0% response (L, low control). Data were normalized according to the following formula:% data = 100 × (data − L)/(H − L)(1)

Statistical analyses were performed with GraphPad Prism v8.0 software (GraphPad Software, Inc. La Jolla, CA, USA). Concentration-response fittings were performed using the four-parameter logistic equation and plotted as mean ± standard error of the mean (SEM). Additional data were shown as dot plots with medians (center line). Since the normal distribution of data, checked by the Kolmogorov–Smirnov test, could not be rejected, statistical comparisons between groups were performed by Student’s *t*-test for unpaired data. Comparisons between more than two groups were performed by one-way analysis of variance (ANOVA), followed by the Tukey’s multiple comparison *post hoc* test or by the nonparametric Kruskal–Wallis test, followed by Dunn’s multiple comparison test for non-normally distributed data. For all experiments, *p* < 0.05 was considered statistically significant. Statistical tests were two-tailed. The need for nonparametric analysis was assessed by the Shapiro–Wilk normality test (alpha = 0.05).

## 5. Conclusions

Exposure to endogenous or exogeneous substances acting as agonists or PAMs at NMDARs may dysregulate calcium currents, with downstream consequences that depend on the cells and circuits affected. Here, we show that REL-1017 may counteract the increases in [Ca^2+^]_in_ caused by endogenous and exogenous molecules in the presence and absence of glutamate. REL-1017 may counteract different mechanisms of QA neurotoxicity, including QA-induced increases in ambient glutamate and the direct agonist effect at the glutamate site of NMDARs. These experiments confirm that the GluN1-2D subtype is the most sensitive to low-concentration glutamate (0.04 μM and 0.2 μM), suggesting that uncompetitive NMDAR blockers with a preference for this subtype, such as REL-1017, may be of particular therapeutic relevance.

## Figures and Tables

**Figure 1 pharmaceuticals-15-00882-f001:**
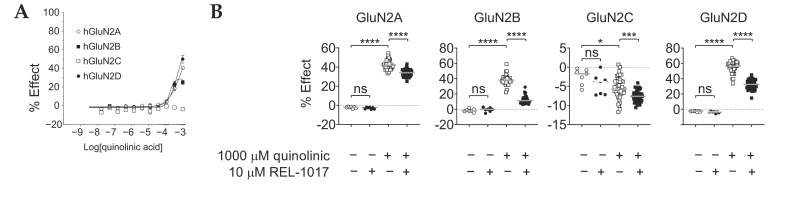
(**A**) Quinolinic acid (QA) is a subtype-selective N-methyl-D-aspartate receptor (NMDAR) partial agonist. QA concentration–response curve (CRC) was performed in the absence of L-glutamate by fluorometric imaging plate reader (FLIPR) calcium assay, using the Chinese hamster ovary (CHO) cell lines expressing the indicated heterodimeric NMDAR. QA, at the two highest tested concentrations, 333 µM and 1000 µM, increased intracellular calcium ([Ca^2+^]_in_) in GluN2A-, GluN2B-, and GluN2D-expressing cell lines, with estimated an EC_50_ of 850 µM, 170 µM, and 520 µM for GluN2A, GluN2B, and GluN2D receptors, respectively. Estimated maximal asymptotic response was 75%, 28%, and 71% of that evoked by 10 μM L-glutamate for GluN2A, GluN2B, and GluN2D receptors, respectively. GluN2C-expressing cells were not activated by any tested concentration of QA. Data are mean ± standard error of the mean (SEM), *n* = 6 wells per group, and were fitted using four-parametric logistic equation with GraphPad Prism v8.0. (**B**) It was shown that 10 µM REL-1017 reduced calcium entry in the presence of 1000 μM QA. QA 1000 μM increased [Ca^2+^]_in_ up to 41%, 37%, and 55% in GluN2A, GluN2B, and GluN2D NMDAR subtypes, respectively. In addition, 10 μM REL-1017 significantly (*p* < 0.0001) decreased these values to 34%, 12%, and 32% in GluN1-GluN2A, GluN1-GluN2B, and GluN1-GluN2D NMDAR subtypes, respectively. In the GluN1-GluN2C cell line, [Ca^2+^]_in_ was −5.5% and −7.6% after 1000 μM QA and 1000 μM QA plus 10 μM REL-1017, suggesting a reduction in baseline NMDAR-mediated [Ca^2+^]_in_, possibly due to an endogenous low-level L-glutamate release by the CHO cell line. Data (*n* = 7 wells for buffer and REL-1017 groups, *n* = 42 wells for the remaining groups) are shown as scatter dot plot, and median value is indicated. **** is *p* < 0.0001; *** is *p* < 0.001; * is *p* < 0.05; ns is not statistically significant.

**Figure 2 pharmaceuticals-15-00882-f002:**
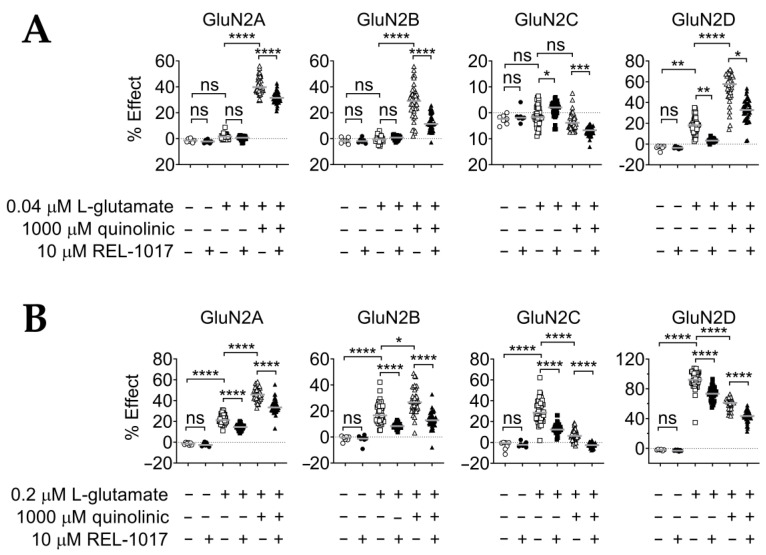
(**A**) Cell lines incubated with 10 µM REL-1017 reduced calcium entry in the presence of 1000 μM QA and 0.04 μM L-glutamate. Then, 1000 μM QA was tested in combination with 0.04 μM L-glutamate; 0.04 μM L-glutamate alone did not elicit [Ca^2+^]_in_ changes above an average of 2% in GluN2A, GluN2B, and GluN2C. However, 0.04 μM L-glutamate increased [Ca^2+^]_in_ by an 18% average in the GluN2D cell line (*p* < 0.01). QA 1000 μM significantly (*p* < 0.0001) increased [Ca^2+^]_in_ in the presence of 0.04 μM L-glutamate in all cell lines, except for the NMDAR GluN2C cell line. In addition, 10 μM REL-1017 significantly decreased [Ca^2+^]_in_ elicited by 1000 μM QA plus 0.04 μM L-glutamate in all cell lines (*p* < 0.0001 for GluN2A and GluN2B; *p* < 0.001 for GluN2C; *p* < 0.05 for GluN2D). Data (*n* = 7 wells for buffer and REL-1017 groups, *n* = 42 wells for the remaining groups) are shown as scatter dot plot, and median value is indicated. (**B**) It was shown that 10 µM REL-1017 reduced calcium entry in the presence of 1000 μM QA and 0.2 μM L-glutamate. Next, 1000 μM QA was tested in combination with 0.2 μM L-glutamate; 0.2 μM L-glutamate alone elicited an average [Ca^2+^]_in_ of 22%, 18%, 30%, and 92% in GluN2A, GluN2B, GluN2C, and GluN2D cell lines, respectively. QA 1000 μM significantly increased [Ca^2+^]_in_ in the presence of 0.2 μM L-glutamate only in GluN2A- or GluN2B-expressing cell lines (*p* < 0.0001 and *p* < 0.05, respectively). In contrast, 1000 μM QA significantly (*p* < 0.0001) decreased [Ca^2+^]_in_ in the presence of 0.2 μM L-glutamate in GluN2C- and GluN2D-expressing cell lines. In addition, 10 μM REL-1017 significantly (*p* < 0.0001) decreased [Ca^2+^]_in_ elicited by 1000 μM QA plus 0.2 μM L-glutamate in all cell lines, in agreement with its NMDAR channel blocking activity. Data (*n* = 7 wells for buffer and REL-1017 groups, *n* = 42 wells for the remaining groups) are shown as scatter dot plot, and median value is indicated. **** is *p* < 0.0001; *** is *p* < 0.001; ** is *p* < 0.01; * is *p* < 0.05; ns is not statistically significant.

**Figure 3 pharmaceuticals-15-00882-f003:**
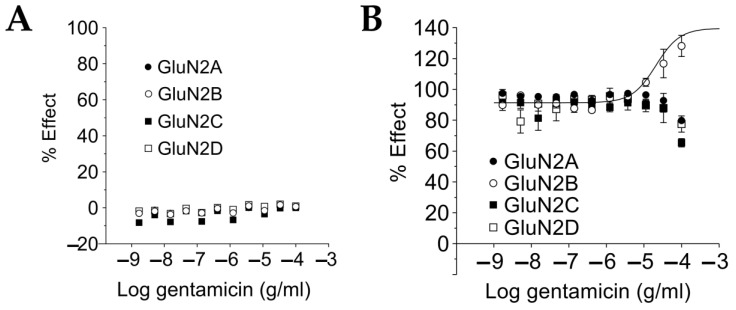
(**A**) Gentamicin had no NMDAR agonist effect. Gentamicin CRC was performed in absence of L-glutamate by FLIPR calcium assay, using the CHO cell lines expressing the indicated heterodimeric NMDAR. Gentamicin did not increase [Ca^2+^]_in_ at any tested concentration and in any cell line, indicating that it cannot ne substituted for L-glutamate in the activation of NMDARs. Data are mean ± SEM, *n* = 6. (**B**) Gentamicin was a positive modulator only at GluN1-GluN2B subtypes in the presence of 10 μM L-glutamate. A second gentamicin CRC was run in similar FLIPR calcium assay conditions as those for the experiment illustrated in the text, but in the presence of 10 μM L-glutamate. Gentamicin increased the [Ca^2+^]_in_ only in the cell line expressing the GluN1-GluN2B subtype, eliciting a maximal response of 128% at 100 µg/mL gentamicin, with a calculated EC_50_ of 21 µg/mL. Data are mean ± SEM, *n* = 6.

**Figure 4 pharmaceuticals-15-00882-f004:**
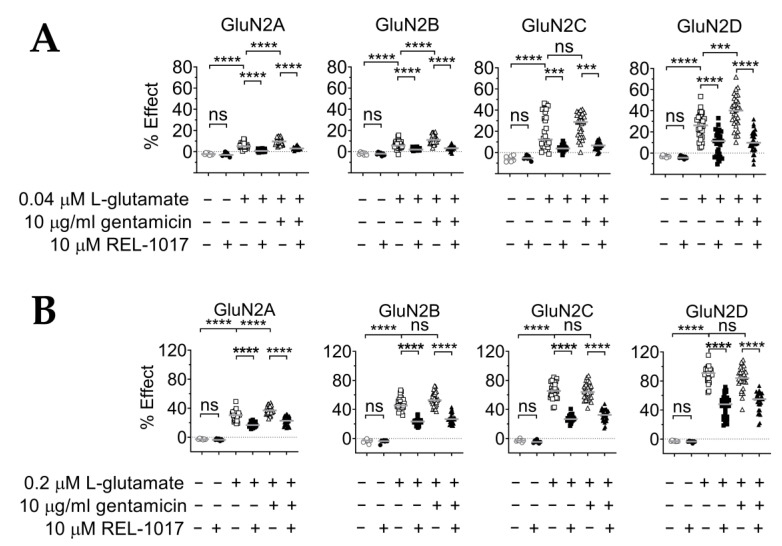
(**A**) Gentamicin was a positive modulator at GluN1-GluN2A, GluN1-GluN2B, and GluN1-GluN2D subtypes, in the presence of 0.04 μM L-glutamate, and REL-1017 reduced [Ca^2+^]_in_ in the presence of 0.04 μM L-glutamate, with or without 10 μM gentamicin, in all tested NMDARs. In addition, 10 μM gentamicin was tested in combination with 0.04 μM L-glutamate; 10 μM gentamicin significantly increased [Ca^2+^]_in_ in all cell lines expressing GluN1-GluN2A, GluN1-GluN2B, or GluN1-GluN2D subtypes (*p* < 0.0001, *p* < 0.0001, and *p* < 0.001, respectively). REL-1017 significantly decreased [Ca^2+^]_in_ elicited by 0.04 μM L-glutamate and 0.04 μM L-glutamate plus 10 μM gentamicin in all cell lines (*p* < 0.0001 in all cell lines and with *p* < 0.001 in GluN1-GluN2C). Data (*n* = 7 wells for buffer and REL-1017 groups, *n* = 30 wells for the remaining groups) are shown as scatter dot plot, and median value is indicated. (**B**) Gentamicin was a positive modulator at the GluN1-GluN2A NMDAR subtype, in the presence of 0.2 μM L-glutamate, and REL-1017 reduced [Ca^2+^]_in_ in the presence of 0.2 μM L-glutamate, with or without 10 μM gentamicin, in all cell lines. Furthermore, 10 μM gentamicin was tested in combination with 0.2 μM L-glutamate; 10 μM gentamicin significantly (*p* < 0.0001) increased [Ca^2+^]_in_ only in cell lines expressing GluN1-GluN2A subtypes. REL-1017 significantly (*p* < 0.0001) decreased [Ca^2+^]_in_ elicited by 0.2 μM L-glutamate and 0.2 μM L-glutamate plus 10 μM gentamicin in all cell lines. Data (*n* = 7 wells for buffer and REL-1017 groups, *n* = 30 wells for the remaining groups) are shown as scatter dot plot, and median value is indicated. **** is *p* < 0.0001; *** is *p* < 0.001; ns, not statistically significant.

**Figure 5 pharmaceuticals-15-00882-f005:**
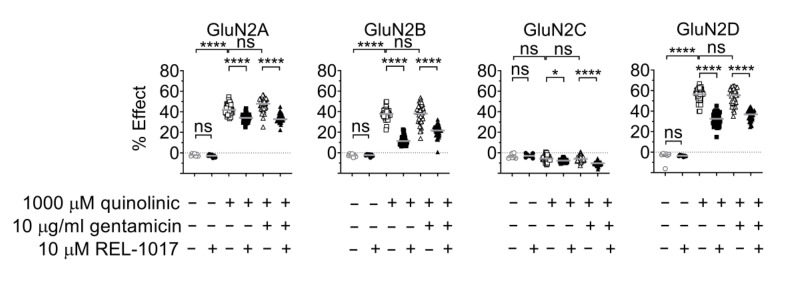
The 1000 μM QA was a partial agonist at GluN1-GluN2A, GluN1-GluN2B, GluN1-GluN2C, or GluN1-GluN2D subtypes and 10 µg/mL gentamicin did not increase [Ca^2+^]_in_ in the presence of 1000 μM QA. REL-1017 significantly decreased [Ca^2+^]_in_ elicited by 1000 μM QA or 1000 μM QA plus 10 μM gentamicin. Data (*n* = 7 wells for buffer and REL-1017 groups, *n* = 42 wells for the remaining groups) are shown as scatter dot plot, and median value is indicated. **** is *p* < 0.0001; * is *p* < 0.05; ns, not statistically significant.

**Table 1 pharmaceuticals-15-00882-t001:** Statistical significance of values reported in Figure 1.

Multiple Comparisons Test	Tukey’s GluN2A *p*-Value	Tukey’s GluN2B *p*-Value	Tukey’s GluN2C *p*-Value	Tukey’s GluN2D *p*-Value
Buffer vs. 10 μM REL-1017	0.9851	0.9851	0.3757	0.9882
Buffer vs. 1000 μM QA	<0.0001	<0.0001	0.0103	<0.0001
Buffer vs. 1000 μM QA + 10 μM REL-1017	<0.0001	<0.0001	<0.0001	<0.0001
10 μM REL-1017 vs. 1000 μM QA	<0.0001	<0.0001	0.7045	<0.0001
10 μM REL-1017 vs. 1000 μM QA + 10 μM REL-1017	<0.0001	<0.0001	0.0054	<0.0001
1000 μM QA vs. 1000 μM QA + 10 μM REL-1017	<0.0001	<0.0001	0.0002	<0.0001
Fisher value F and degrees of freedom dF1 and dF2 [3,92]	442	266	14	319

Data passed Shapiro–Wilk normality test (alpha = 0.05), and statistical analysis was performed with analysis of variance (ANOVA) test (Fisher value F and degrees of freedom dF1 and dF2 were F [3,93] = 442, F [3,92] = 266, F [3,93] = 14, and F [3,94] = 319 for GluN2A, GluN2B, GluN2C, and GluN2D groups, respectively; *p*-value: 0.0001 for all groups) followed by Tukey’s multiple comparison test (93, 92, 93, and 94 degrees of freedom for GluN2A, GluN2B, GluN2C, and GluN2D groups, respectively).

**Table 2 pharmaceuticals-15-00882-t002:** Statistical significance of values reported in Figure 2A.

Multiple Comparisons Test	Tukey’s GluN2A *p*-Value	Tukey’s GluN2B *p*-Value	Dunn’s GluN2C *p*-Value	Dunn’s GluN2D *p*-Value
Buffer vs. 10 μM REL-1017	0.9991	>0.9999	>0.9999	>0.9999
Buffer vs. 0.04 μM L-glu	0.4072	0.9992	>0.9999	0.0088
0.04 μM L-glu vs. 0.04 μM L-glu + 10 μM REL-1017	0.7378	0.9878	0.0410	0.0014
0.04 μM L-glu vs. 0.04 μM L-glu + 1000 μM QA	<0.0001	<0.0001	0.5041	<0.0001
0.04 μM L-glu + 1000 μM QA vs. 0.04 μM L-glu + 1000 μM QA + 10 μM REL-1017	<0.0001	<0.0001	0.0002	0.0189
Fisher value F and degrees of freedom dF1 and dF2 [5,176]	775	94		

GluN2A and GluN2B data, which passed Shapiro–Wilk normality tests (alpha = 0.05), were analyzed with ANOVA (Fisher value F and degrees of freedom dF1 and dF2 were F [5,176] = 775 and F [5,172] = 94 for GluN2A and GluN2B groups, respectively; *p*-value: 0.0001 for both groups), followed by Tukey’s multiple comparison test (176 and 172 degrees of freedom for GluN2A and GluN2B groups, respectively). GluN2C and GluN2D data, which did not pass the Shapiro–Wilk normality test (alpha = 0.05), were analyzed by nonparametric Kruskal–Wallis test, followed by Dunn’s multiple comparison test.

**Table 3 pharmaceuticals-15-00882-t003:** Statistical significance of values reported in Figure 2B.

Multiple Comparisons Test	Tukey’s GluN2A *p*-Value	Dunn’s GluN2B *p*-Value	Tukey’s GluN2C *p*-Value	Tukey’s GluN2D *p*-Value
Buffer vs. 10 μM REL-1017	0.9998	>0.9999	0.9992	>0.9999
Buffer vs. 0.2 μM L-glu	<0.0001	<0.0001	<0.0001	<0.0001
0.2 μM L-glu vs. 0.2 μM L-glu + 10 μM REL-1017	<0.0001	<0.0001	<0.0001	<0.0001
0.2 μM L-glu vs. 0.2 μM L-glu + 1000 μM QA	<0.0001	0.0358	<0.0001	<0.0001
0.2 μM L-glu + 1000 μM QA vs. 0.2 μM L-glu + 1000 μM QA + 10 μM REL-1017	<0.0001	<0.0001	<0.0001	<0.0001
Fisher value F and degrees of freedom dF1 and dF2 [5,176]	307		128	275

Data, which passed Shapiro–Wilk normality tests (alpha = 0.05), were analyzed with ANOVA (Fisher value F and degrees of freedom dF1 and dF2 were F [5,176] = 307, F [5,174] = 128, and F [5,171] = 275 for GluN2A, GluN2C, and GluN2D groups, respectively; *p*-value: 0.0001 for all groups), followed by Tukey’s multiple comparison test. GluN2B data, which did not pass Shapiro–Wilk normality tests (alpha = 0.05), were analyzed by nonparametric Kruskal–Wallis test, followed by Dunn’s multiple comparison test.

**Table 4 pharmaceuticals-15-00882-t004:** Statistical significance of values reported in Figure 4A.

Multiple Comparisons Test	Tukey’s GluN2A *p*-Value	Tukey’s GluN2B *p*-Value	Dunn’s GluN2C *p*-Value	Tukey’s GluN2D *p*-Value
Buffer vs. 10 μM REL-1017	0.9916	>0.9999	>0.9999	>0.9999
Buffer vs. 0.04 μM L-glu	<0.0001	<0.0001	<0.0001	<0.0001
0.04 μM L-glu vs. 0.04 μM L-glu + 10 μM REL-1017	<0.0001	<0.0001	0.0006	<0.0001
0.04 μM L-glu vs. 0.04 μM L-glu + 10 μg/mL gentamicin	<0.0001	<0.0001	0.6481	0.0002
0.04 μM L-glu + 10 μg/mL gentamicin vs. 0.04 μM L-glu + 10 μg/mL gentamicin + 10 μM REL-1017	<0.0001	<0.0001	0.0002	<0.0001
Fisher value F and degrees of freedom dF1 and dF2 [5,176]	87	55		38

GluN2A, GluN2B, and GluN2D data, which passed Shapiro–Wilk normality tests (alpha = 0.05), were analyzed with ANOVA (Fisher value F and degrees of freedom dF1 and dF2 were F [5,128] = 87, F [5,126] = 55, and F [5,125] = 38 for GluN2A, GluN2B, and GluN2D groups, respectively; *p*-value: 0.0001 for all groups), followed by Tukey’s multiple comparison test, while GluN2C data, which did not pass Shapiro–Wilk normality tests (alpha = 0.05), were analyzed by nonparametric Kruskal–Wallis test, followed by Dunn’s multiple comparison test.

**Table 5 pharmaceuticals-15-00882-t005:** Statistical significance of values reported in Figure 4B.

Multiple Comparisons Test	Tukey’s GluN2A *p*-Value	Dunn’s GluN2B *p*-Value	Tukey’s GluN2C *p*-Value	Dunn’s GluN2D *p*-Value
Buffer vs. 10 μM REL-1017	0.9998	>0.9999	0.9996	>0.9999
Buffer vs. 0.2 μM L-glu	<0.0001	<0.0001	<0.0001	<0.0001
0.2 μM L-glu vs. 0.2 μM L-glu + 10 μM REL-1017	<0.0001	<0.0001	<0.0001	<0.0001
0.2 μM L-glu vs. 0.2 μM L-glu + 10 μg/mL gentamicin	<0.0001	>0.9999	0.9951	>0.9999
0.2 μM L-glu + 10 μg/mL gentamicin vs. 0.2 μM L-glu + 10 μg/mL gentamicin + 10 μM REL-1017	<0.0001	<0.0001	<0.0001	<0.0001
Fisher value F and degrees of freedom dF1 and dF2 [5,176]	139		186	

GluN2A and GluN2C data, which passed Shapiro–Wilk normality tests (alpha = 0.05), were analyzed with ANOVA (Fisher value F and degrees of freedom dF1 and dF2 were F [5,128] = 139 and F [5,127] = 186 for GluN2A and GluN2C groups, respectively; *p*-value: 0.0001 for all groups), followed by Tukey’s multiple comparison test, while GluN2B and GluN2D data, which did not pass the normality test, were analyzed by nonparametric Kruskal–Wallis test, followed by Dunn’s multiple comparison test.

**Table 6 pharmaceuticals-15-00882-t006:** Statistical significance of values reported in Figure 5.

Multiple Comparisons Test	Dunn‘s GluN2A *p*-Value	Dunn‘s GluN2B *p*-Value	Dunn‘s GluN2C *p*-Value	Dunn‘s GluN2D *p*-Value
Buffer vs. 10 μM REL-1017	>0.9999	>0.9999	>0.9999	>0.9999
Buffer vs. 1000 μM QA	<0.0001	<0.0001	>0.9999	<0.0001
1000 μM QA vs. 1000 μM QA + 10 μM REL-1017	<0.0001	<0.0001	0.0125	<0.0001
1000 μM QA vs. 1000 μM QA + 10 μg/mL gentamicin	0.8007	>0.9999	>0.9999	>0.9999
1000 μM QA + 10 μg/mL gentamicin vs. 1000 μM QA + 10 μg/mL gentamicin + 10 μM REL-1017	<0.0001	<0.0001	<0.0001	<0.0001

Data did not pass Shapiro–Wilk normality tests (alpha = 0.05) and were analyzed by the nonparametric Kruskal–Wallis test, followed by Dunn’s multiple comparison test.

## Data Availability

Data is contained within the article.

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
