# Peer review of "The N-Methyl-D-Aspartate Receptor Blocker REL-1017 (Esmethadone) Reduces Calcium Influx Induced by Glutamate, Quinolinic Acid, and Gentamicin"

_pharmaceuticals, 2022, doi:10.3390/ph15070882_

Round 1

Reviewer 1 Report

The manuscript entitled The N-Methyl-D-Aspartate Receptor Blocker REL-1017 (Esmethadone) Reduces Calcium Influx Induced by Glutamate, Quinolinic Acid, and Gentamicin described the results of a well-designed and performed experiment focusing on the mechanism of action of REL-1017, a novel drug candidate, which is in early phase clinical studies. 

The results of this study are important to understand how REL-1017 can improve neuropsychiatric diseases and some evidence suggested that NMDA receptors are targeted. The manuscript is well-written and it is suitable to be published in Pharmaceuticals journal.

Some minor observations:

1. The resolution of Figure 1, 2, and 4 should be improved. It can hardly be deciphered.

2. To be easier to understand, the F values reported in the footnote of Table 1, 2, 3, 4, 5, and 6 should be included in the table by placing them in the specified group.

3. Conclusions - The first paragraph should be more concise. Some of the sentences can be moved to discussion, and the conclusions should be rephrased to be more specific  

Reviewer 2 Report

This very interesting manuscript is devoted to investigation a role of gentamicin in increase NMDAR-mediated [Ca2+]in in recombinant cell lines expressing heterodimeric NMDARs both in the presence and absence of glutamate and the ability of REL-1017 to antagonize the effects of gentamicin on NMDARs activated by glutamate and the ability of REL-1017 to antagonize the effects of gentamicin and quinolinic acid.

Obtained results show that these data may improve our understanding of the mechanism of action of REL-1017, including its potential to protect cells from excessive calcium entry via

NMDA receptors hyperactivated by glutamate and endogenous or exogenous substances, such as quinolinic acid and gentamicin.

There is a one moment which has to be explained. It was shown, that REL-1017 reduced [Ca2+]in induced by the addition of 0.04 or 0.02 µM L-glutamate in all NMDAR cell lines in the absence of gentamicin. Whether means it that the glutamate in concentration 0.04 or 0.02 µM is the same modulator of NMDA receptors and  induces  increase[Ca2+]in in itself?

There is a typo on the page number12.
